# Graphical analysis of guideline adherence to detect systemwide anomalies in HIV diagnostic testing

Ronald George Hauser[1,2]*, Ankur Bhargava[1,3], Cynthia A. Brandt[1,3], Maggie Chartier[4], Marissa M. Maier[5,6]

1 Veterans Affairs Connecticut Healthcare System, West Haven, CT, United States of America,
2 Department of Laboratory Medicine, Yale University School of Medicine, New Haven, CT, United States of America, 3 Department of Emergency Medicine, Yale University School of Medicine, New Haven, CT, United States of America, 4 Office of Specialty Care Services, Veterans Health Administration, Washington, DC, United States of America, 5 Veterans Affairs Portland Health Care System, Portland, OR, United States of America, 6 Division of Infectious Diseases, Oregon Health and Sciences University, Portland, OR, United States of America

* ronald.hauser@yale.edu

**Data Availability Statement:** Due to US Department of Veterans Affairs (VA) regulations and our ethics agreements, the analytic data sets used for this study are not permitted to leave the

## Abstract

### Background

Analyses of electronic medical databases often compare clinical practice to guideline recommendations. These analyses have a limited ability to simultaneously evaluate many interconnected medical decisions. We aimed to overcome this limitation with an alternative method and apply it to the diagnostic workup of HIV, where misuse can contribute to HIV transmission, delay care, and incur unnecessary costs.

### Methods

We used graph theory to assess patterns of HIV diagnostic testing in a national healthcare system. We modeled the HIV diagnostic testing guidelines as a directed graph. Each node in the graph represented a test, and the edges pointed from one test to the next in chronological order. We then graphed each patient's HIV testing. This set of patient-level graphs was aggregated into a single graph. Finally, we compared the two graphs, the first representing the recommended approach to HIV diagnostic testing and the second representing the observed patterns of HIV testing, to assess for clinical practice deviations.

### Results

The HIV diagnostic testing of 1.643 million patients provided 8.790 million HIV diagnostic test results for analysis. Significant deviations from recommended practice were found including the use of HIV resistance tests (n = 3,007) and HIV nucleic acid tests (n = 16,567) instead of the recommended HIV screen.

VA firewall without a Data Use Agreement. This limitation is consistent with other studies based on VA data. However, VA data are made freely available to researchers with an approved VA study protocol. For more information, please visit https://www.virec.research.va.gov or contact the VA Information Resource Center at VIReC@va.gov.

**Funding:** The authors received no specific funding for this work.

**Competing interests:** The authors have declared that no competing interests exist.

## Conclusions

We developed a method that modeled a complex medical scenario as a directed graph. When applied to HIV diagnostic testing, we identified deviations in clinical practice from guideline recommendations. The model enabled the identification of intervention targets and prompted systemwide policy changes to enhance HIV detection.

## Introduction

Observational medical databases allow healthcare systems to electronically measure how well the care of patients conforms to clinical guidelines [1]. The retrospective measurement of clinical guideline adherence previously required manual chart review, a time-consuming and subjective process [2]. Electronic reviews (e.g., database queries) of guideline adherence, in contrast, generally take less time, can be easily modified, repeatedly executed, and scaled to larger sample sizes with minimal additional effort [3–5].

A limitation of electronic reviews as opposed to manual reviews is that they often evaluate less complex medical scenarios [6]. Diverse technologies have attempted to address this limitation. Natural language processing (NLP) can convert critical data elements and context-specific medical decisions into a structured form [7, 8]. Guidelines Interchange Format (GLIF), Guidelines Element Model (GEM) and other projects express complex clinical guidelines in a computable format [9–12]. Even with these tools, the analysis of complex medical scenarios continues to present a challenge.

Decision pathways are a method to diagram a complex medical scenario, such as the Center for Disease Control and Prevention's (CDC) HIV Diagnostic Testing Guidelines, which recommend a hierarchical sequence of testing to diagnose HIV [13]. Rather than assess adherence to each decision in the hierarchy, authors typically review a single decision point. For example, Cane et. al. reviewed HIV resistance testing in patients with a low HIV viral load [14]. Improved analysis methods would promote the review of an entire medical scenario, rather than a single decision, for adherence to medical guidelines.

Graph theory provides a mathematical construct capable of modeling complex decision pathways [9, 15]. A graph, according to graph theory, consists of nodes, commonly represented as circles, connected by edges, commonly represented as lines. Graphs in which the edges denote a path to be followed are termed directed graphs and their edges are represented as arrows. Attributes of the graph can represent additional information: a line's style can represent proper or improper adherence to current guidelines (e.g., solid line for adherence, dashed line for non-adherence), and the thickness of the line can represent the utilization frequency of an edge.

To facilitate the use of observational medical databases to evaluate guideline adherence in a complex medical scenario, a method involving graph theory is introduced. Using graph theory, we created a model of the CDC's HIV Diagnostic Testing Guidelines to evaluate the adherence of clinicians to HIV diagnostic testing recommendations provided by the CDC, identify intervention targets, and suggest an appropriate intervention strategy.

## Methods

We developed a method to assess adherence to HIV testing guidelines in a large healthcare system by leveraging historical electronic health record (EHR) data. We assessed the CDC's HIV

Diagnostic Testing Guidelines by modeling them as a directed graph. For each patient we created a directed graph of their HIV testing. A patient's graph begins at the start node and travels along the graph's edges to reach the node of the next test performed in chronological order. The set of patient-level graphs were aggregated into a summary graph of all HIV testing sequences performed within our healthcare system. Finally, we compared these two graphs, the first representing the CDC's recommended approach to HIV diagnostic testing and the second graph representing the testing patterns we found in our healthcare system. The comparison allowed us to assess for deviations from the recommended guidelines. The Supplement and online code repository contain more technical details (e.g., step-by-step examples, data structures, algorithms) (S1-S6 Figs in S1 File, S1-S7 Tables in S1 File) [16].

## Modeling of HIV diagnostic testing guidelines as a graph

**Recommended approach to HIV diagnostic tests.** The CDC publishes guidelines for HIV Diagnostic Testing [13]. They recommend an HIV-1/2 antigen/antibody combination immunoassay to screen for HIV followed by a confirmation test, an HIV-1/HIV-2 antibody differentiation immunoassay. They recommend following a negative or indeterminate confirmation test with an HIV-1 nucleic acid test (NAT). The CDC no longer recommends the HIV-1 Western blot, a test previously used for diagnosis. Except in unusual circumstances, only patients with confirmed HIV should receive HIV resistance tests or HIV NAT. (See S8 Table in S1 File for additional details about HIV diagnostic tests).

**Representation of guidelines as a graph.** As a first step to measuring adherence, the guidelines were modeled as a graph (Fig 1A). Nodes in the graph represented either an HIV test or the result of a specific HIV test, such as an HIV resistance test or a positive HIV screen, respectively. Edges in the graph connected tests recommended to be performed in sequence, with edges pointing from one test in the sequence to the next. For example, an edge connects a positive HIV screen node with a positive HIV confirmation node. The graphical model of the guidelines balanced the intent of the guidelines with modifications to simplify the measurement of adherence.

The graph intentionally contains a start node and an end node. The edge from the start node to the HIV screen node conveys the guideline's recommendation to perform this test first. Similarly, the edges to the end node denote which tests the guideline considers appropriate to stop the diagnostic workup. After a negative HIV screen, for example, it is appropriate stop the HIV workup, so an edge exists between the negative HIV screen node and the end node. In contrast, a patient lost to follow-up after a positive HIV screen would not adhere to the guidelines because they would benefit from an HIV confirmation test. The graph conveys this by the absence of an edge between a positive HIV screen and the end node. The inclusion of the start and end nodes allows the graph to contain important edges used later in the analysis.

HIV resistance tests are not diagnostic for HIV, but they are still included in the graph. After an HIV diagnosis is confirmed, the guidelines recommend the use of HIV resistance tests [17]. This explains the edge from a positive HIV confirmation to an HIV resistance test. Resistance tests are not a recommended part of HIV diagnosis, but researchers have documented its inappropriate utilization [14]. To differentiate between appropriate and inappropriate utilization of the HIV resistance test, we included it, and the edges denoting adherence to the guidelines, in the graph.

The graph permits repeated HIV diagnostic workups. For example, a patient with a negative HIV screen may, perhaps after a potential exposure, undergo a second HIV screen, also with a negative result. The graph models this scenario by a "loop", a special type of edge that points to

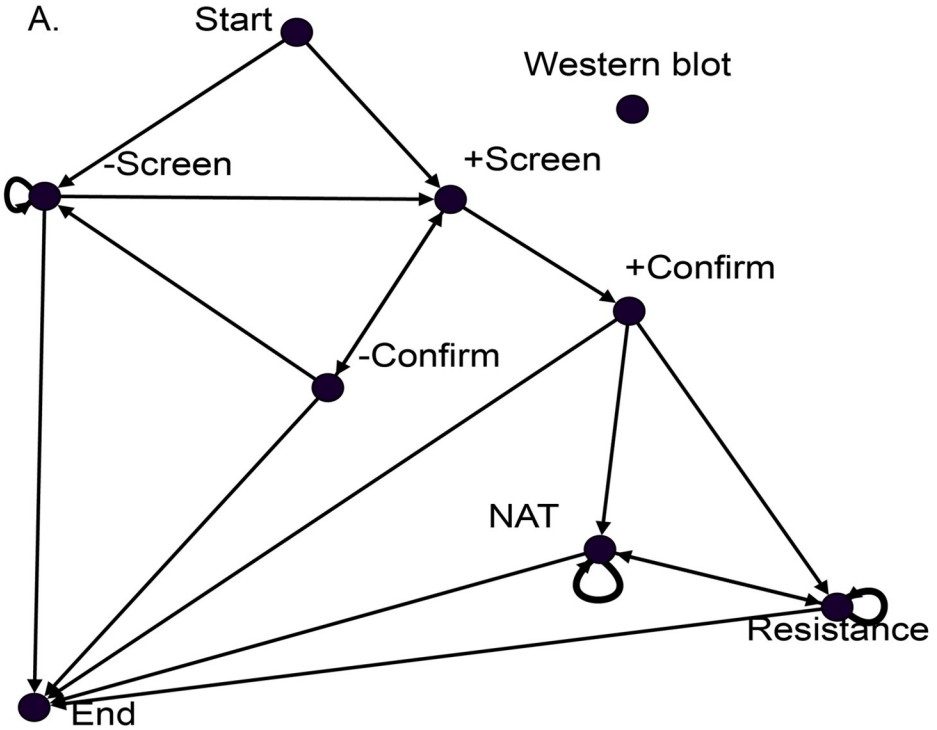

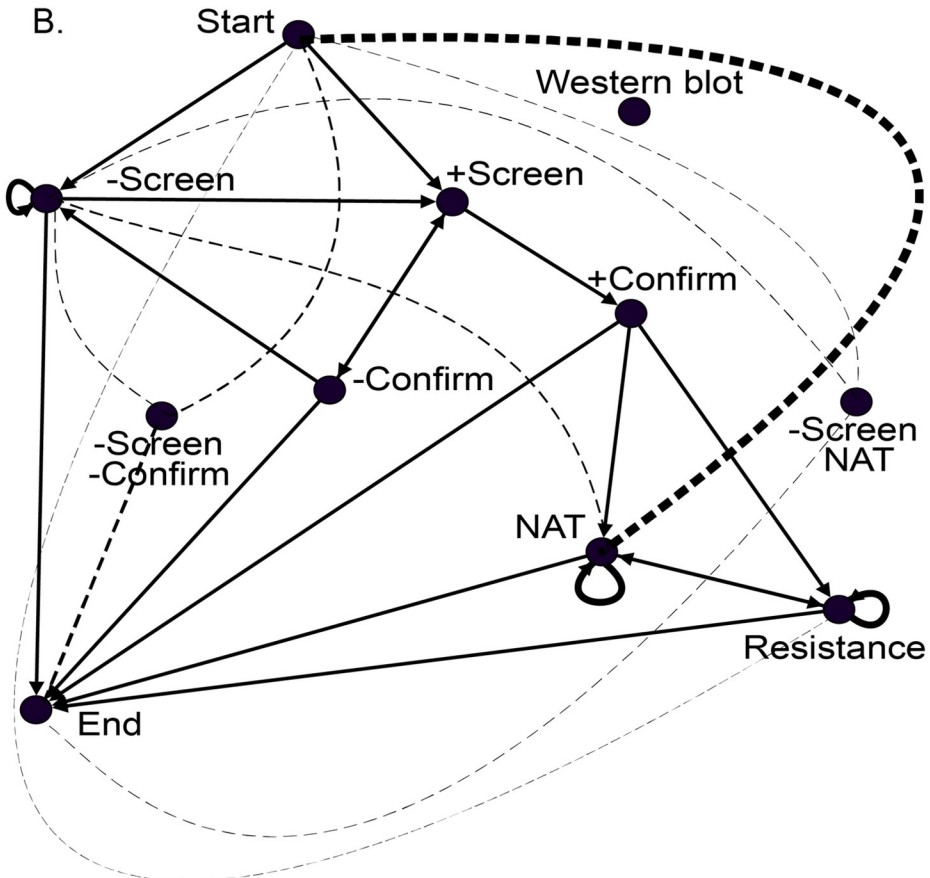

**Fig 1. A.** Modeling of HIV Diagnostic Testing Guidelines as a Graph. Arrows connect HIV tests that should occur chronologically according to guidelines. The arrow points from the first test to the second. **B.** Nonadherence to HIV Diagnostic Testing Guidelines as a Graph. Solid lines denote observed adherence to guidelines (See Fig 1A). Dashed lines denote observed nonadherence to guidelines. The graph shows the 11 nonadherent edges with at least 1,000 observations, which comprise 84% (54,149/64,405) of the total nonadherent observations. Line thickness of dashed lines denotes the number of nonadherent tests. -, the test result is negative; +, the test result is positive; Screen, HIV-1/2 antigen/antibody combination immunoassay; confirm, HIV-1/HIV-2 antibody differentiation immunoassay; NAT, HIV-1 nucleic acid test (NAT); Resistance, HIV resistance test.

the same node from where it originated. A loop can be found at the negative HIV screen node in the graph (Fig 1A).

## Modeling of patient HIV testing as a graph

**Data source.** Study data originates from the Veterans Health Administration's (VA) Corporate Data Warehouse (CDW), a relational database that aggregates medical data, including laboratory results, from 130 separate healthcare facilities [18]. These healthcare facilities are located across the continental United States, Alaska, Hawaii, and the Philippines. Nearly all facilities contributed identifiable data for the full duration of the study. We identified HIV laboratory tests and standardized their results, including checks for manual data entry errors, with previously published methods [19, 20].

The VA healthcare system maintains an HIV registry that contains a list of patients with known HIV, including their date of diagnosis. The HIV registry helped define the study population.

**Study population.** The study population consists of patients that underwent HIV diagnostic testing at VA facilities between January 2015 to January 2019 inclusive (Fig 2). We excluded patients with known HIV because diagnostic testing guidelines did not apply to them. For example, a patient with known HIV may appropriately receive a test for HIV resistance at their first visit to our healthcare system.

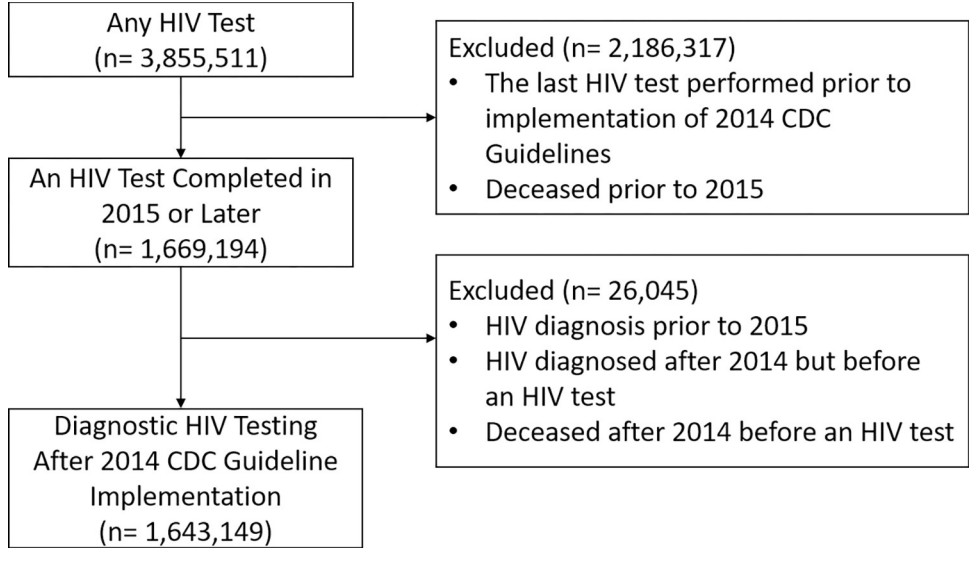

**Fig 2. Determination of study population.**

### Comparison of patient HIV testing to guidelines

**Data analysis.**   To evaluate the adherence of HIV testing to guidelines, we assembled a directed graph to summarize the sequential HIV testing performed on patients within our healthcare system. We began by arranging the HIV tests performed for each patient in chronological order. Next, the chronologically arranged HIV tests were converted to edges, where the edge began at the first test and pointed to the next. Each patient had an edge from the start node to the first HIV test they received. Likewise, each patient had an edge from the last HIV test they received to the end node. The list of edges was aggregated to count the number of occurrences of each edge (e.g., an edge from the start node to negative HIV screen node occurred 10 times). The output, a table of edges and a count of their occurrence, was converted to a directed graph.

To determine adherence to the HIV Diagnostic Testing Guidelines, each edge was classified as adherent or nonadherent to the guidelines. We denoted the adherence of an edge to the guidelines by line style. An edge drawn with a solid line represented adherence, while an edge drawn with a dashed line represented nonadherence.

The source code (i.e., C#, T-SQL) to conduct the analysis is available online [16]. Gephi and Inkscape were used to draw the graphs. S7 Fig in S1 File shows the data flow and manipulation.

**Determination of significant findings.**   To verify examples of nonadherence to the guidelines discovered through the graphical model, we manually reviewed patient charts. Fifty patients were reviewed for each type of nonadherence. Disagreements were adjudicated by HIV subject matter expert consensus. Inter-annotator agreement was reported by Cohen's kappa.

### Consideration of absolute time

To construct a clinically meaningful model of guideline adherence with a directed graph, an important consideration is absolute time, in addition to the chronological sequence of tests (e.g., test 1 → test 2). By absolute time we mean the elapsed time measured in, for example, seconds or minutes, between one test and another. We modified the graphical model of guideline adherence to account for absolute time and improve its clinical interpretation. These considerations are detailed in the S1 File.

### Cost estimation

The Center for Medicare and Medicaid Services' (CMS) 2018 Clinical Laboratory Fee Schedule provides costs for HIV tests [21]. The cost of nonadherent tests was determined by counting the number of individual tests which the graph identified as nonadherent to the guidelines.

## Results

### Study population description

Over 3.855 million patients underwent HIV testing in our healthcare system between 1999 and 2019 (Fig 2). The initial years (1999–2014) were used to determine patients with HIV prior to the as part of the registry of HIV patients. The later years (2015–2019), contained 1.643 million patients who underwent HIV diagnostic testing after our healthcare system implemented the 2014 CDC Guidelines. The demographics of these patients are included in Table 1. These patients received care, including 8.790 million HIV diagnostic test results, at 130 facilities.

**Table 1. Study population demographics.** The population totaled 1,643,149 patients.

| | |
|---|---|
| Age in years: Mean (+/- SD) | 52.2 (±17.2) |
| Sex (% Male) | 80.2% |
| Race (% White) | 57.5% |
| Ethnicity (% Non-Hispanic) | 75.8% |

## Assessment of adherence to HIV testing guidelines

The graphical analysis of the study population's HIV testing produced 331 unique edges (14 adherent, 317 nonadherent). Many of the edges occurred infrequently with only 14 edges (3 adherent, 11 nonadherent) having over 1,000 occurrences (Fig 1B). On review of the nonadherent edges by test (e.g., HIV NAT), we found three recurring scenarios: (1) HIV NAT with or without an HIV screen, (2) HIV resistance testing used in place of an HIV screen, and (3) the performance of a confirmation test after a negative HIV screen. Cohen's kappa for interrater agreement of adherence to guidelines was 0.78 (97% agreement; 97/100).

   **Scenario 1: HIV nucleic acid tests (NAT).** On manual chart review of the nonadherent edges involving HIV NATs, we found the majority (86%, 43/50) represented true nonadherence because the HIV NAT was used in combination or in lieu of the HIV screen. Specifically, we did not find evidence to sufficiently explain the utilization of HIV NATs such as (1) the appropriate use of HIV NAT to diagnose acute HIV, (2) HIV NAT as a follow-up to a negative HIV confirmation test, or (3) HIV NAT performed in a patient with existing HIV. Of the 11 nonadherent edges with over 1,000 observations in the graph, 7 edges involved the HIV NAT. These edges had 23,728 occurrences from orders placed by 9,927 clinicians representing all 130 facilities.

   **Scenario 2: HIV resistance test used in place of an HIV screen.** On review of the nonadherent edges involving HIV resistance tests, we also found the majority (100%, 50/50) represented true nonadherence. The patients reviewed did not have existing HIV, as recommended by the guidelines. The one nonadherent edge with over 1,000 observations in the graph pointed from the start node to the HIV resistance node. This indicates an HIV resistance test was the first HIV test performed in these 1,644 patients. A total of 56% (73/130) of facilities had at least one observation of this edge. A few facilities accounted for 61% (1002/1644) of the observations, and clinicians who placed these orders could be identified within these facilities. When these clinicians were contacted by phone, they erroneously believed the HIV resistance test was the HIV screen and agreed to modify their HIV test ordering practices.

   **Scenario 3: combined HIV screen and confirmation tests.** Review of patient's medical charts with an HIV screen and confirmation test performed together revealed many of these patients had received a 5th generation HIV screen. This test, in contrast to the 4th generation test, combines the HIV screen and HIV confirmation into a single test. The analysis classified this scenario as nonadherent because we did not anticipate it prior to our review of patient charts, but after review, we believe it represents adherence to guidelines.

## The cost of guideline nonadherence

We identified the total number of nonadherent HIV tests: 16,567 (86% of 19,264) HIV NAT and 3,007 (100% of 3,007) HIV resistance tests. The CMS cost per test was $94.55 for the HIV NAT and $286.05 for the HIV resistance test. In total, the estimated cost of nonadherent testing was $2.427 million in 2018 United States dollars.

## Discussion

This graphical model uses a directed graph to model guidelines, an idea shared with the Guide-Line Interchange Format (GLIF) and others [9, 22]. It also relies heavily on temporal relationship, which other authors have studied in depth in a medical context [23]. Unlike previous graphical methods, our graphical model (1) evaluates guideline adherence within a population instead of at the individual level, (2) assesses nonadherent clinical practice, in addition to adherent clinical practice, and (3) quantifies the impact of the observed nonadherence while identifying targets for intervention.

The examples of nonadherence from our analysis convey important lessons for reviews of guidelines adherence. First, the strategy to reverse nonadherence may originate from the review itself. Nonadherence limited to relatively few facilities or clinicians suggests a simple intervention (e.g., phone call) to reverse course, such as with nonadherent HIV resistance tests (Figs 3 and 4). Nonadherence to HIV NAT affected a larger proportion of the health system and required a more intensive intervention (e.g., systemwide campaign). Second, the availability of structured data does not obviate the need for manual chart review. Through manual chart review, we identified facilities utilizing the $5^{th}$ generation HIV test, which we did not consider in the model.

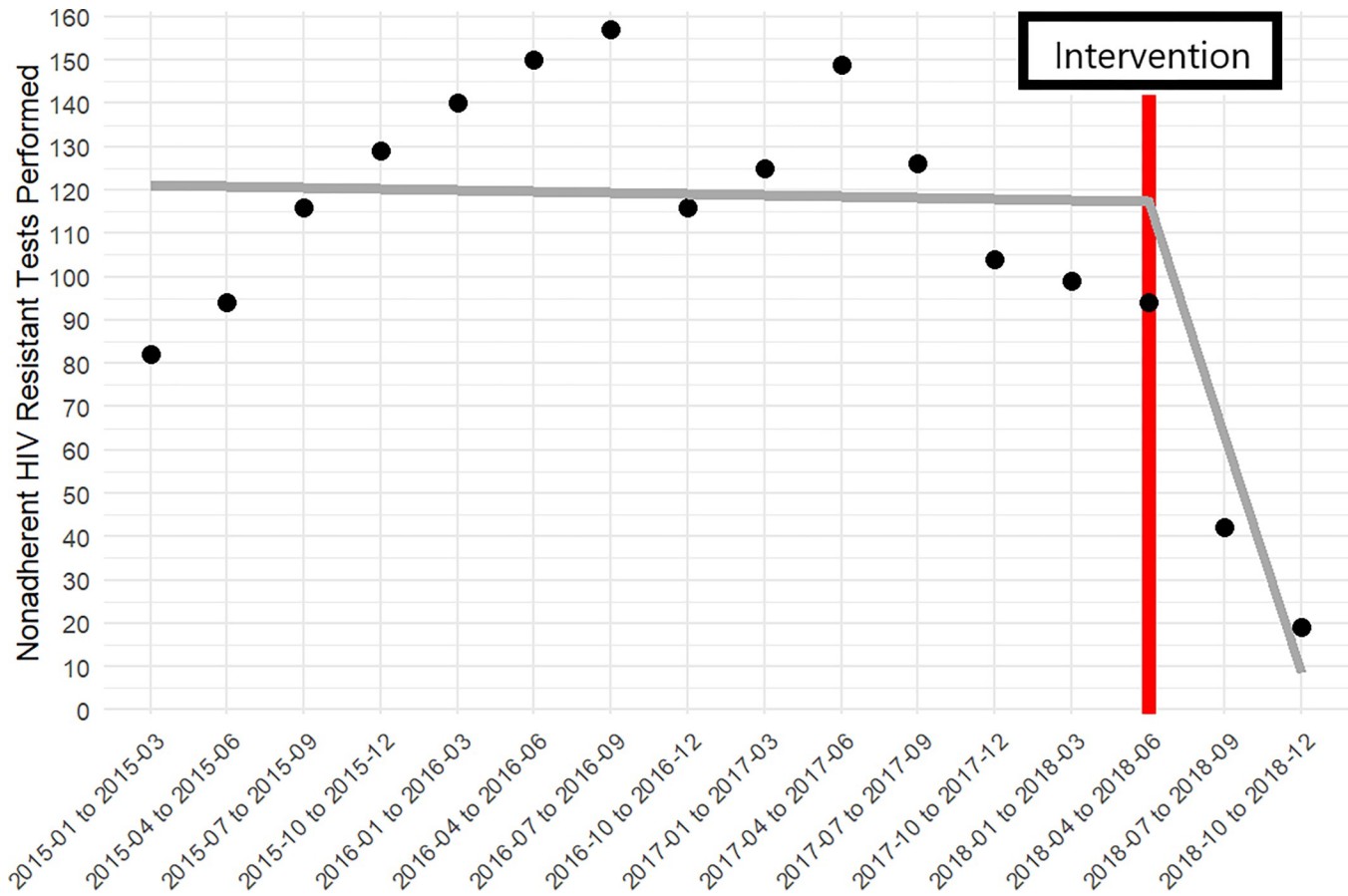

**Fig 3. Reduction of inappropriate HIV resistance tests performed at outlier facilities before and after an intervention (i.e., phone calls).** Total tests performed at outlier facilities before and after an intervention (gray; continuous piecewise linear spline). The red vertical line represents the timing of the intervention.

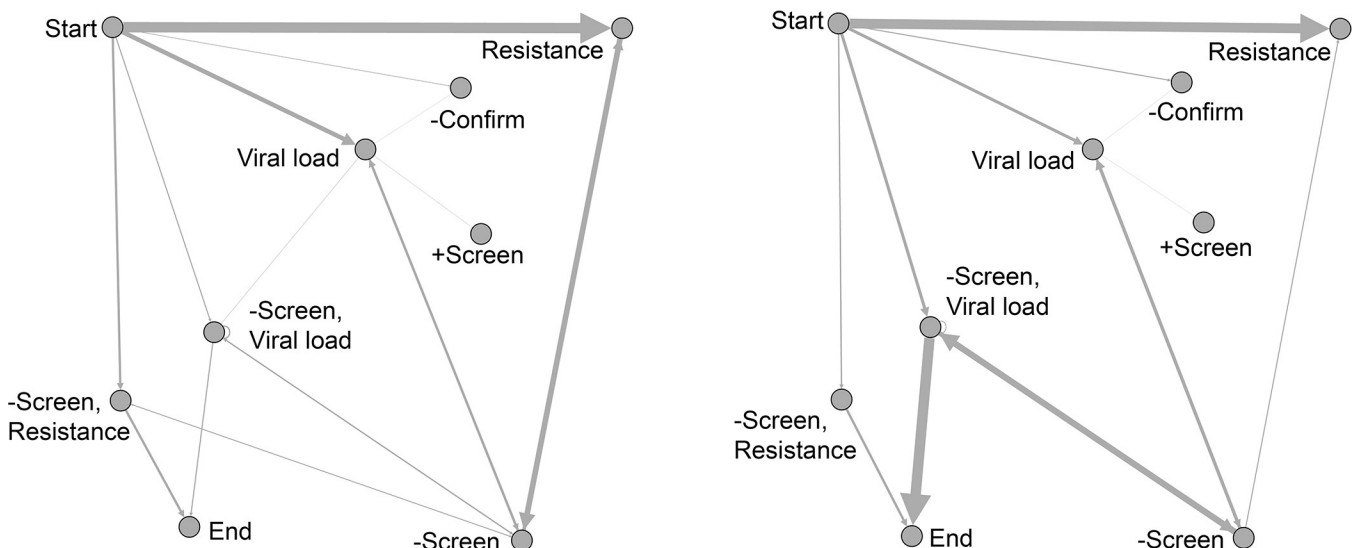

**Fig 4. Reduction of inappropriate HIV resistance tests performed at outlier facilities.** The most common shared edges are shown (A) before (A) and after (B) the intervention.

### Strengths of modeling guideline adherence as a graph

Modeling guideline adherence as a graph has multiple strengths. First, the graph is easy to understand. The method does not involve advanced mathematics (e.g., algebra, calculus) [24]. Clinicians generally appreciate the graph as a model of the expected (Fig 1A) and observed (Fig 1B) pattern of diagnostic testing for HIV after a brief orientation.

Second, although simple to understand, the graph can represent a complex process. Non-recommended diagnostic tests (e.g., HIV Western blot), in addition to the recommended tests, are included in the graph of expected HIV diagnostic testing. It also conveys inappropriate tests to start and stop a diagnostic workup. For example, an HIV diagnostic workup should not end after a positive HIV screen, an HIV confirmation is needed. Finally, the graph models repeated workups, such that a patient could become HIV positive after a negative HIV screen because an arrow points to a positive HIV screen from a negative HIV screen (Fig 1A).

Third, the process scales easily to large populations. Our healthcare system is the largest integrated healthcare system in the United States, and we conducted this analysis on a commodity desktop PC. Most applications will not require expensive computer hardware and may be repeatedly run as part of a plan-do-check-act (PDCA) cycle to support iterative quality improvement. Even as it scales to large populations, it remains highly specific, identifying individual clinicians who performed inappropriate HIV tests in a healthcare system with over 10,000 clinicians.

### Limitations of modeling guideline adherence as a graph

We encountered certain difficulties when modeling guideline adherence as a graph. First, we had to balance graph accuracy with interpretability. For example, the consideration of absolute time created additional nodes and edges, increasing the complexity of the graph. (See S1 File–Consideration of Absolute Time.) The incorporation of absolute time increased the model's accuracy, so we tolerated its increased complexity. As a second example, we chose to exclude indeterminate HIV confirmation results from the model because they happened too rarely to provide a benefit.

Second, the development of graphical guideline adherence models may require iterative revision. On review of the current model, we became aware of 5[th] generation HIV testing, which performs the HIV screen and confirm in a single test. The current HIV diagnostic guidelines describe the sequential performance of an HIV confirmation only after a positive HIV screen. To distinguish between 5[th] generation HIV testing and the incorrect performance of an HIV confirmation after a negative HIV screen, the model would require a revision (i.e., a new node to represent a 5[th] generation HIV test).

In the future, we plan to build an interface between this data analysis method and a standardized observational data model [25].

## Conclusion

We developed a graphical model to determine if complex medical scenarios adhered to established guidelines. The model applies to patient populations, rather than individuals. With an observational database as the input, we demonstrated the method via an electronic, retrospective review of HIV diagnostic testing in over one million patients. The method identified >20,000 occurrences of inappropriate utilization of the HIV NAT test and HIV resistance tests, which cost an estimated $2.427 million dollars. This led to systemwide changes in policy to reduce nonadherent orders and enhance detection of HIV. This approach is in no way specific to HIV and may be applied to diverse medical scenarios.

## Supporting information

**S1 File. Graphical analysis of guideline adherence to detect systemwide anomalies in HIV diagnostic testing.**
(DOCX)

## Acknowledgments

We would like to acknowledge Joanna Moran, who helped us obtain data related to the HIV registry.

## Author Contributions

**Conceptualization:** Ronald George Hauser.

**Data curation:** Marissa M. Maier.

**Project administration:** Cynthia A. Brandt.

**Resources:** Maggie Chartier, Marissa M. Maier.

**Supervision:** Ronald George Hauser.

**Visualization:** Ankur Bhargava.

**Writing – original draft:** Ronald George Hauser, Ankur Bhargava, Maggie Chartier, Marissa M. Maier.

**Writing – review & editing:** Ronald George Hauser, Ankur Bhargava, Cynthia A. Brandt, Maggie Chartier, Marissa M. Maier.

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
