## [Decision Letter · Decision Letter 0]

14 Dec 2021

PONE-D-20-27454Graphical Analysis of Guideline Adherence to Detect Systemwide Anomalies in HIV Diagnostic TestingPLOS ONE

Dear Dr. Hauser,

Thank you for submitting your manuscript to PLOS ONE. After careful consideration, we feel that it has merit but does not fully meet PLOS ONE’s publication criteria as it currently stands. Therefore, we invite you to submit a revised version of the manuscript that addresses the points raised during the review process. I would like to sincerely apologize for the delay you have incurred with your submission. It has been exceptionally difficult to secure reviewers to evaluate your study. We have now received two completed reviews; their comments are available below. The reviewers have raised significant scientific concerns about the study that need to be addressed in a revision.

Please revise the manuscript to address all the reviewer's comments in a point-by-point response in order to ensure it is meeting the journal's publication criteria. Please note that the revised manuscript will need to undergo further review, we thus cannot at this point anticipate the outcome of the evaluation process.

We look forward to receiving your revised manuscript.

Kind regards,

Miquel Vall-llosera Camps

Senior Editor

PLOS ONE

Journal Requirements:

2. Thank you for including the following ethics statement on the submission details page:

'N/A - This work is exempt from human subjects review'

in the ethics statement in the Methods section of your manuscript, please specify whether the medical data used for your study is anonymized before access.

3. Thank you for stating the following financial disclosure: "NO - The funders had no role in study design, data collection and analysis, decision to publish, or preparation of the manuscript."

Reviewers' comments:

Reviewer's Responses to Questions

**Comments to the Author**

1. Is the manuscript technically sound, and do the data support the conclusions?

Reviewer #1: Yes

Reviewer #2: Yes

2. Has the statistical analysis been performed appropriately and rigorously? 

Reviewer #1: Yes

Reviewer #2: N/A

3. Have the authors made all data underlying the findings in their manuscript fully available?

Reviewer #1: No

Reviewer #2: No

4. Is the manuscript presented in an intelligible fashion and written in standard English?

Reviewer #1: Yes

Reviewer #2: Yes

5. Review Comments to the Author

Reviewer #1: The authors propose to use graphical models to evaluate if guidelines are followed. This is done by comparing the observed and guideline recommended graphs for HIV diagnostic testing. My comments are:

a) I think similar procedures are in place at many places. For example, when my group works with EHRs we perform similar procedures to check for data quality. Without user friendly software being available that allows easy implementation of the method is different settings, the impact of this is unclear.

b) In EHRs it is pretty common to have entry errors. Did the manual chart review check for those?

c) Guidelines are not laws and are meant to guide but not completely dictate clinical decision making. Sometimes external circumstances justify not following the guidelines. Do the authors have any sense of how often the clinicians made a conscious decision not to follow the guidelines?

d) It would be useful and interesting to have more discussion of the downstream effect of these checks. For example, more details on the intervention and how it was implemented in this setting would be helpful.

e) The authors state “The method does not involve advanced mathematics (e.g., algebra, calculus).” Can you give some examples of methods that require that?

Reviewer #2: This is a nice paper, clearly presented, good, generally reusable methodology. My only comment is that you miss an opportunity to show off your approach. Why not do the following:

1) Have Figure 1B show potential erroneous edges (or maybe anticipate the important ones and just incliude those, if there are too many possible permutations).

2) Show the figure currently in 1B as a separate figure in the Results section and use line thinkness to show frequency, as you mention in your introduction.

3) Add a new two-part figure to complement the current figure 3 that shows before and after, as weighted-edge graphs.

It is not often I ask for a chance to re-review a paper (maybe never) but in this case I would be delighted to see a revised paper with the graphs as described above.

6. PLOS authors have the option to publish the peer review history of their article (what does this mean?). If published, this will include your full peer review and any attached files.

Reviewer #1: No

Reviewer #2: No

---

## [Author Response · Author response to Decision Letter 0]

24 Jan 2022

Reviewer #1: The authors propose to use graphical models to evaluate if guidelines are followed.

This is done by comparing the observed and guideline recommended graphs for HIV diagnostic testing. My comments are:

a) I think similar procedures are in place at many places. For example, when my group works with EHRs we perform similar procedures to check for data quality. Without user friendly software being available that allows easy implementation of the method is different settings, the impact of this is unclear.

Response a) We believe the reviewer has suggested that procedures similar to what we have described are “in place at many places”. We agree that there are similar processes for visual checking of data quality and that the availability of user-friendly software for easy access to these procedures is indeed lacking. But we believe that the packaging of the data quality checks with guidelines in a graphical form is novel, and we have made the software available in the public domain for others to use. 

b) In EHRs it is pretty common to have entry errors. Did the manual chart review check for those?

Response b) Data entry errors are certainly common when manual processes are used. The dataset for this project was created by automated platforms used by clinical laboratories. Typographic errors are relatively rare with these automated platforms. (See https://pubmed.ncbi.nlm.nih.gov/28505339/.) We employed manual chart review to determine the clinical scenario in which the test was used.

c) Guidelines are not laws and are meant to guide but not completely dictate clinical decision making. Sometimes external circumstances justify not following the guidelines. Do the authors have any sense of how often the clinicians made a conscious decision not to follow the guidelines?

Response c) We agree that guidelines should not dictate the practice of medicine. We found that the use of HIV resistance tests instead of the HIV screen (Scenario 2) were 100% (n=50) nonadherent to guidelines and mistakes by these clinicians. In the discussion we mention a “simple intervention (e.g., phone call) to reverse course, such as with nonadherent HIV resistance tests (Figure 3).” We did indeed place a phone call to these providers, and they freely admitted they had mistakenly ordered the HIV resistance test instead of the HIV screen test. This was likely the reason for the drastic improvement in Figure 3.

d) It would be useful and interesting to have more discussion of the downstream effect of these checks. For example, more details on the intervention and how it was implemented in this setting would be helpful.

Response d) We do not have more details about the intervention. As described in the response above, the intervention consisted of us making providers aware of their erroneous orders over the phone. The extent of this intervention is described in the discussion. We also do not want to distract from the novelty of the paper, mainly the graphical methodology.

e) The authors state “The method does not involve advanced mathematics (e.g., algebra, calculus).” Can you give some examples of methods that require that?

Response e) Test utilization methods that involve advanced mathematics would include this paper written by one of the study authors (see https://www.ncbi.nlm.nih.gov/pmc/articles/PMC4355837/). The appendix of this paper contains an algebraic derivation.

Reviewer #2: This is a nice paper, clearly presented, good, generally reusable methodology. My only comment is that you miss an opportunity to show off your approach. Why not do the following:

1) Have Figure 1B show potential erroneous edges (or maybe anticipate the important ones and just include those, if there are too many possible permutations).

Response 1) We appreciate the kind comments from this reviewer, and we certainly do want to show off our approach! The reviewer has asked we show the erroneous edges in Figure 1B. These solid edges in this graph show patterns that adhere to guidelines. The dotted lines show nonadherent patterns.

2) Show the figure currently in 1B as a separate figure in the Results section and use line thickness to show frequency, as you mention in your introduction.

Response 2) We appreciate the suggestion, and we created Figure 1B as requested. As stated in the paper, we reviewed 8.790 million HIV tests with an estimated 20 thousand incorrect tests performed. The ratio of correct tests to incorrect tests makes the line thickness for the correctly performed tests much thicker than those of the incorrect tests. The line thickness of the incorrect tests is smaller and less noticeable to the reader. We want to draw the reader’s attention to the incorrect test ordering, because even though these tests make up a minority, they could lead to a missed diagnosis of HIV. We want to highlight this. For these reasons, we did not include the figure in the manuscript. 

3) Add a new two-part figure to complement the current figure 3 that shows before and after, as weighted-edge graphs.

Response 3) Weighted-edge graphs use the line thickness to convey information about the frequency of the edge. We interpreted this question in a similar line of reasoning to the question above, which requested the use of line thickness, as in a weighted-edge graph. We came to a similar conclusion when deciding to use a weighted-edge graph in this situation. Because the number of incorrect tests makes a thinner line compared to the thicker line of the correctly performed tests, the reader’s attention is drawn away from the incorrectly performed tests. Even though we do not feel this suggestion is in the best interest of the manuscript, we appreciate the reviewer’s thoughtful critique.

It is not often I ask for a chance to re-review a paper (maybe never) but in this case I would be delighted to see a revised paper with the graphs as described above.

---

## [Decision Letter · Decision Letter 1]

6 May 2022

PONE-D-20-27454R1Graphical Analysis of Guideline Adherence to Detect Systemwide Anomalies in HIV Diagnostic TestingPLOS ONE

Dear Dr. Hauser,

Thank you for submitting your manuscript to PLOS ONE. After careful consideration, we feel that it has merit but does not fully meet PLOS ONE’s publication criteria as it currently stands. Therefore, we invite you to submit a revised version of the manuscript that addresses the points raised during the review process.

While both reviewers now endorse publication, we note that your revised manuscript does not appear to incorporate your responses to the reviewers' comments. As these queries were raised by the reviewers, it is likely that readers of your manuscript will have similar questions. As such, we request that you further revise your manuscript to incorporate your responses to the reviewers (though regarding comment a) from Reviewer #1 we note that the software used in this study is already available via reference 16). As these are few minor textual changes and the substance of the work has been approved for publication by both reviewers, I do not expect further review will be necessary once these changes have been made.

We look forward to receiving your revised manuscript.

Kind regards,

Hugh Cowley

Staff Editor

PLOS ONE

Journal Requirements:

Reviewers' comments:

Reviewer's Responses to Questions

**Comments to the Author**

1. If the authors have adequately addressed your comments raised in a previous round of review and you feel that this manuscript is now acceptable for publication, you may indicate that here to bypass the “Comments to the Author” section, enter your conflict of interest statement in the “Confidential to Editor” section, and submit your "Accept" recommendation.

Reviewer #1: All comments have been addressed

Reviewer #2: (No Response)

2. Is the manuscript technically sound, and do the data support the conclusions?

Reviewer #1: Yes

Reviewer #2: Yes

3. Has the statistical analysis been performed appropriately and rigorously? 

Reviewer #1: Yes

Reviewer #2: N/A

4. Have the authors made all data underlying the findings in their manuscript fully available?

Reviewer #1: Yes

Reviewer #2: Yes

5. Is the manuscript presented in an intelligible fashion and written in standard English?

Reviewer #1: Yes

Reviewer #2: Yes

6. Review Comments to the Author

Reviewer #1: All comments have been addressed

Reviewer #2: It appears that the authors have responded to Reviewer #1's comments by disputing them in their response letter, rather than modifying the paper to address the issues, one way or another, so that future readers with the same concerns can seee their reponses *in the paper* (not in some online review thread. I will let Reviewer #1 respond about the appropriateeness, but I feel it is imporant to point out.

The authors have provided only a minimum response to my previous comments. They have modified Figure 1B as requested (thank you) but have left the figure in the Methods section, rather than make it a new figure in the Results section (as requested). Their refusal to add weight to the lines seems disingenuous they could certainly use a non-linear scale (for example a log scale) to show relative counts through varying line thickness. Instead, they turned down an opportunity to improve the display of their work.

A word about my recommendation: I don't feel "accept" is appropriate, given the almost complete lack of response to the reviewer comments, while "minor revision" seems inappropriate because they have ignored the opportunity to make the reevisions the first time, and "major revision" is too severe for what is requested. I choose "reject" to get the edtior's attention because my real response to the editor is "I did my best to help them improve the paper an they basically ignored me so I see no further need to be involved in the review. It's a good paper. Too bad they don't want ot make it a great paper. now it is up to you".

7. PLOS authors have the option to publish the peer review history of their article (what does this mean?). If published, this will include your full peer review and any attached files.

Reviewer #1: No

Reviewer #2: No

---

## [Author Response · Author response to Decision Letter 1]

27 May 2022

Response 

To the editor:

We appreciate your perseverance in providing a through review of this paper, balancing the various viewpoints from reviewers and authors. To simplify this hopefully final review, we have done our best to include all reviewer comments and suggestions in the manuscript. The first section is a point-by-point review of the previous round of comments noting where the changes in the manuscript occurred. The second section is the response to the last round of comments. Our most recent comments are labeled with "(5/2022)" to distinguish them from previous comments.

Section 1: Editor

From the editor: While both reviewers now endorse publication, we note that your revised manuscript does not appear to incorporate your responses to the reviewers' comments. As these queries were raised by the reviewers, it is likely that readers of your manuscript will have similar questions. As such, we request that you further revise your manuscript to incorporate your responses to the reviewers (though regarding comment a) from Reviewer #1 we note that the software used in this study is already available via reference 16). As these are few minor textual changes and the substance of the work has been approved for publication by both reviewers, I do not expect further review will be necessary once these changes have been made.

Response (5/2022): We have copied our initial responses to the reviewers here, highlighting additional changes we have made to the manuscript to incorporate the responses to the reviewers. 

To the editor:

Thank you for addressing the delay incurred with this submission. We appreciate that it has been difficult to secure reviewers to evaluate the study.

We have reviewed the style requirements, amending the manuscript as necessary. 

1. Changed headings to sentence case

2. Modified heading style 1, 2, and 3 to match the recommendations

3. Added a funding section: “Funding: The study had no designated funds, and the authors received no specific funding for this work. Therefore, funders had no role in study design, data collection and analysis, decision to publish, or preparation of the manuscript.”

4. Modified the figure references from “Fig.” to “Fig”.

5. Specified the data was identifiable when we received it. See line 142, “Nearly all facilities contributed identifiable data for the full duration of the study.” Note: Due to the sensitive nature of HIV and the VA data sharing policy, we are not permitted to place this data set in the public domain.

Reviewer #1: The authors propose to use graphical models to evaluate if guidelines are followed.

This is done by comparing the observed and guideline recommended graphs for HIV diagnostic testing. My comments are:

a) I think similar procedures are in place at many places. For example, when my group works with EHRs we perform similar procedures to check for data quality. Without user friendly software being available that allows easy implementation of the method is different settings, the impact of this is unclear.

Response a) We believe the reviewer has suggested that procedures similar to what we have described are “in place at many places”. We agree that there are similar processes for visual checking of data quality and that the availability of user-friendly software for easy access to these procedures is indeed lacking. But we believe that the packaging of the data quality checks with guidelines in a graphical form is novel, and we have made the software available in the public domain for others to use. 

Response a (5/2022)) This questions was incorporated in the previous edit as the editor noted in their comments, “though regarding comment a) from Reviewer #1 we note that the software used in this study is already available via reference 16”. We did not further modify the manuscript in response to this question.

b) In EHRs it is pretty common to have entry errors. Did the manual chart review check for those?

Response b) Data entry errors are certainly common when manual processes are used. The dataset for this project was created by automated platforms used by clinical laboratories. Typographic errors are relatively rare with these automated platforms. (See https://pubmed.ncbi.nlm.nih.gov/28505339/.) We employed manual chart review to determine the clinical scenario in which the test was used.

Response b (5/2022)) We incorporated this concern into the manuscript. Specifically, we updated the description of the test result standardization process to note that the process would identify data entry errors. “We identified HIV laboratory tests and standardized their results, including checks for manual data entry errors, with previously published methods.19,20”

c) Guidelines are not laws and are meant to guide but not completely dictate clinical decision making. Sometimes external circumstances justify not following the guidelines. Do the authors have any sense of how often the clinicians made a conscious decision not to follow the guidelines?

Response c) We agree that guidelines should not dictate the practice of medicine. We found that the use of HIV resistance tests instead of the HIV screen (Scenario 2) were 100% (n=50) nonadherent to guidelines and mistakes by these clinicians. In the discussion we mention a “simple intervention (e.g., phone call) to reverse course, such as with nonadherent HIV resistance tests (Figure 3).” We did indeed place a phone call to these providers, and they freely admitted they had mistakenly ordered the HIV resistance test instead of the HIV screen test. This was likely the reason for the drastic improvement in Figure 3.

Response c (5/2022)) We have added the following sentence to better explain why these clinicians did make a conscious decision not to follow the HIV testing guidelines. “When these clinicians were contacted by phone, they erroneously believed the HIV resistance test was the HIV screen and agreed to modify their HIV test ordering practices.” See Result > “Scenario 2: HIV resistance test used in place of an HIV screen”.

d) It would be useful and interesting to have more discussion of the downstream effect of these checks. For example, more details on the intervention and how it was implemented in this setting would be helpful.

Response d) We do not have more details about the intervention. As described in the response above, the intervention consisted of us making providers aware of their erroneous orders over the phone. The extent of this intervention is described in the discussion. We also do not want to distract from the novelty of the paper, mainly the graphical methodology.

Response d (5/2022)) The paper contains a mention of our intervention in three separate locations: methods, discussion, and Figure 3 caption. We added the description of the intervention in the results with this revision. 

• Results: “When these clinicians were contacted by phone, they erroneously believed the HIV resistance test was the HIV screen and agreed to modify their HIV test ordering practices.”

• Discussion: “First, the strategy to reverse nonadherence may originate from the review itself. Nonadherence limited to relatively few facilities or clinicians suggests a simple intervention (e.g., phone call) to reverse course, such as with nonadherent HIV resistance tests (Fig 3).”

• Figure 3 caption: “Reduction of Inappropriate HIV Resistance Tests Performed at Outlier Facilities Before and After an Intervention (i.e., phone calls).” 

e) The authors state “The method does not involve advanced mathematics (e.g., algebra, calculus).” Can you give some examples of methods that require that?

Response e) Test utilization methods that involve advanced mathematics would include this paper written by one of the study authors (see https://www.ncbi.nlm.nih.gov/pmc/articles/PMC4355837/). The appendix of this paper contains an algebraic derivation.

Response e (5/2022)) We added a reference to this paper alongside the sentence quoted above. “The method does not involve advanced mathematics (e.g., algebra24, calculus).”

Reviewer #2: This is a nice paper, clearly presented, good, generally reusable methodology. My only comment is that you miss an opportunity to show off your approach. Why not do the following:

1) Have Figure 1B show potential erroneous edges (or maybe anticipate the important ones and just include those, if there are too many possible permutations).

Response 1) We appreciate the kind comments from this reviewer, and we certainly do want to show off our approach! The reviewer has asked we show the erroneous edges in Figure 1B. These solid edges in this graph show patterns that adhere to guidelines. The dotted lines show nonadherent patterns.

Response 1 (5/2022)) We made this change as requested by Review 2. (Figure 1B, dashed lines show nonadherent edges.)

2) Show the figure currently in 1B as a separate figure in the Results section and use line thickness to show frequency, as you mention in your introduction.

Response 2) We appreciate the suggestion, and we created Figure 1B as requested. As stated in the paper, we reviewed 8.790 million HIV tests with an estimated 20 thousand incorrect tests performed. The ratio of correct tests to incorrect tests makes the line thickness for the correctly performed tests much thicker than those of the incorrect tests. The line thickness of the incorrect tests is smaller and less noticeable to the reader. We want to draw the reader’s attention to the incorrect test ordering, because even though these tests make up a minority, they could lead to a missed diagnosis of HIV. We want to highlight this. For these reasons, we did not include the figure in the manuscript. 

Response 2 (5/2022)) We created a new figure as the reviewer suggested. We used the line thickness the reviewer suggested as well. See revised Fig 1B. Rather than apply the line thickness to all lines, we implement the line thickness for only the inappropriate tests. We think this is a compromise between what the reviewer suggested and our concerns about interpretability in our previous response. 

3) Add a new two-part figure to complement the current figure 3 that shows before and after, as weighted-edge graphs.

Response 3) Weighted-edge graphs use the line thickness to convey information about the frequency of the edge. We interpreted this question in a similar line of reasoning to the question above, which requested the use of line thickness, as in a weighted-edge graph. We came to a similar conclusion when deciding to use a weighted-edge graph in this situation. Because the number of incorrect tests makes a thinner line compared to the thicker line of the correctly performed tests, the reader’s attention is drawn away from the incorrectly performed tests. Even though we do not feel this suggestion is in the best interest of the manuscript, we appreciate the reviewer’s thoughtful critique.

Response 3 (5/2022)) A new two-part figure was added to complement Figure 3. It is labeled Figure 4.

It is not often I ask for a chance to re-review a paper (maybe never) but in this case I would be delighted to see a revised paper with the graphs as described above.

These are additional comments from the second round of reviews.

Section 2: Reviewer 2, Q6

6. Review Comments to the Author

Reviewer #1: All comments have been addressed

Reviewer #2: It appears that the authors have responded to Reviewer #1's comments by disputing them in their response letter, rather than modifying the paper to address the issues, one way or another, so that future readers with the same concerns can seee their reponses *in the paper* (not in some online review thread. I will let Reviewer #1 respond about the appropriateeness, but I feel it is imporant to point out.

Response (5/2022)) We have gone through the previous reviews point-by-point above and added content to the paper to explain each point.

1) The authors have provided only a minimum response to my previous comments. They have modified Figure 1B as requested (thank you) but have left the figure in the Methods section, rather than make it a new figure in the Results section (as requested). 

Response (5/2022)) We created a new figure as the reviewer suggested. We used the line thickness the reviewer suggested as well. Rather than apply the line thickness to all lines, we implement the line thickness for only the inappropriate tests. We think this is a compromise between what the reviewer suggested and our concerns about interpretability in our previous response.

2) Their refusal to add weight to the lines seems disingenuous they could certainly use a non-linear scale (for example a log scale) to show relative counts through varying line thickness. Instead, they turned down an opportunity to improve the display of their work.

Response (5/2022)) A new two-part figure was added to complement Figure 3. It is labeled Figure 4.

A word about my recommendation: I don't feel "accept" is appropriate, given the almost complete lack of response to the reviewer comments, while "minor revision" seems inappropriate because they have ignored the opportunity to make the reevisions the first time, and "major revision" is too severe for what is requested. I choose "reject" to get the edtior's attention because my real response to the editor is "I did my best to help them improve the paper an they basically ignored me so I see no further need to be involved in the review. It's a good paper. Too bad they don't want ot make it a great paper. now it is up to you".

---

## [Editor Report · Decision Letter 2]

10 Jun 2022

Graphical Analysis of Guideline Adherence to Detect Systemwide Anomalies in HIV Diagnostic Testing

PONE-D-20-27454R2

Dear Dr. Hauser,

We’re pleased to inform you that your manuscript has been judged scientifically suitable for publication and will be formally accepted for publication once it meets all outstanding technical requirements.

Kind regards,

Hugh Cowley

Staff Editor

PLOS ONE
---

## [Editor Report · Acceptance letter]

24 Jun 2022

PONE-D-20-27454R2 

Graphical Analysis of Guideline Adherence to Detect Systemwide Anomalies in HIV Diagnostic Testing 

Dear Dr. Hauser:

I'm pleased to inform you that your manuscript has been deemed suitable for publication in PLOS ONE. Congratulations! Your manuscript is now with our production department. 

Kind regards, 

on behalf of

Mr Hugh Cowley 

Staff Editor

PLOS ONE